# Risk factors and provider awareness of sexually transmitted enteric pathogens among men who have sex with men

Timothy Isaac Miller,[1] Stephanie Banning,[2] Joshua A. Lieberman[1]

**ABSTRACT** Sexual transmission of enteric pathogens among men who have sex with men (MSM) is well documented, although whether providers are cognizant of this risk when MSM patients present with gastrointestinal symptoms has not been studied. Over 34 months at a major tertiary metropolitan medical system, this study retrospectively analyzed 436 BioFire FilmArray Gastrointestinal results from 361 patients documented as MSM. An extensive chart review was performed, including specific sexual behaviors, socioeconomic risk factors, and whether providers charted a sexual history when a patient presented for care. Overall BioFire positivity rate was 62% with no significant difference in positivity between persons living with HIV and those without. Patients charted as sexually active had a significantly increased odds ratio (OR) of a positive result compared to those who were not. Anilingus had the highest OR. Providers charted any type of sexual history in 40.6% of cases, and HIV/infectious disease providers were significantly more likely to do this compared to other subspecialties. Sexual transmission of enteric pathogens within MSM is ongoing, and patients are at risk regardless of living with HIV. Not all sexual behaviors have the same associated risk, highlighting opportunities to decrease transmission. Increased provider vigilance and better patient education on sexual transmission of enteric pathogens are needed to reduce the disease burden.

**IMPORTANCE** Our work adds several key findings to the growing body of literature describing the epidemiology of enteric pathogens as sexually transmitted infections among men who have sex with men (MSM). We analyzed clinical test results, housing status, provider awareness, sexual behaviors, and symptoms for 361 patients. We found that any sexual activity was associated with an increased risk of diarrheal pathogen detection, whereas being unhoused was not a risk factor. These findings suggest separate transmission networks between unhoused persons, who are also at risk of infectious diarrhea, and MSM. Moreover, our study suggested low awareness among patient-facing clinicians that diarrheal pathogens can be sexually transmitted. Together, our findings indicate an important opportunity to disrupt transmission cycles by educating clinicians on how to assess and counsel MSM patients.

**KEYWORDS** MSM, enteric pathogens, enteric bacteria, gay, homosexual, men who have sex with men, anilingus, BioFire

Address correspondence to Timothy Isaac Miller, timiller@uw.edu.

The authors declare no conflict of interest.

Since the 1970s and 1980s, men who have sex with men (MSM) have been known to be at a higher risk for enteric pathogens (1–6). More recently, there has been an increase in published studies on this topic, likely due to the detection and spread of multidrug-resistant (MDR) enteric pathogens. Several shigellosis outbreaks in MSM communities, including of MDR organisms, have been described globally (7–13). With the more widespread use of whole-genome sequencing, single-nucleotide polymor-

phisms have identified outbreaks of distinct clades of *Shigella* and *Campylobacter* within the MSM community, including the international spread of MDR organisms (14–16).

The increased enteric pathogen risk in MSM is thought to be due to specific sexual behaviors that increase opportunities for fecal–oral transmission. A higher number of sexual partners, bathhouses, group sex, and anilingus have all been associated with a higher risk (17–24). Additionally, MSM patients can be asymptomatic carriers of enteric pathogens, further increasing the risk of spread during sexual contact (25, 26).

Multiplex PCR has emerged as an efficient method to rapidly test for enteric pathogens, typically with higher sensitivity than stool culture (27, 28). This has made it easier to study the risk within MSM, including specific pathogens or serovars not resolved by culture. In recent prior studies from our institution, 268 stool multiplex PCR tests performed on the MSM population in those with gastrointestinal (GI) symptoms found a 56.3% overall positivity rate compared to a general population rate of 33.5% (28, 29). However, these studies did not assess sexual behavioral or socioeconomic risk factors, nor the frequency or quality of sexual history obtained by treating clinicians. To our knowledge, no studies have assessed provider awareness of enteric sexually transmitted infection (STI) risk in the MSM community. Understanding these factors may enable providers to identify the risk to patients and thus better educate and treat them to reduce spread. The main purpose of this study is to further characterize risk factors associated with MSM enteric pathogen acquisition and to assess how often providers take a sexual history when MSM patients present with diarrheal symptoms.

## MATERIALS AND METHODS

### Population

The Laboratory Information System (LIS) was searched for male-identified patients ≥18 years with a BioFire FilmArray Gastrointestintal panel (herein, "Biofire") result between 1 January 2019 and 11 November 2019. Biofire is a rapid multiplex, FDA-approved PCR test performed on fresh stool in Cary Blair medium. The assay tests for 22 different pathogens with a sensitivity of 94.5%–100% and a specificity of 97.1%–100% depending on the target (30). To identify MSM patients, the charts of those patients in the electronic medical record were searched using a custom script for at least one of the following free-text terms: "gay," "homosexual," "MSM," "versatile," "tops," "bottoms," "LGBT," "male partners," "RAI" (receptive anal intercourse), "CRAI" (condomless receptive anal intercourse), or "anal intercourse." Detailed chart review was performed for patients matching the criteria. Patients were excluded if not MSM, did not have male genitalia, testing was performed on an inpatient whose diarrhea started after hospital admission, or the test was performed as a test of cure for a known diagnosis.

### Laboratory data

HIV infection at the time of BioFire testing and the most recent quantitative viral load were also recovered from the LIS and/or patient chart. CD4 count was not analyzed since in many instances the most recent count was >6 months or years prior to stool pathogen testing. Lab protocol included routine, concurrent culture for *Aeromonas* spp. When either *Shigella/Enteroinvasive Escherichia coli* (EIEC) or *Campylobacter* spp. was detected by molecular testing, stool culture and antibiotic susceptibility testing were performed reflexively. Reported antibiotics were ampicillin, trimethoprim/sulfamethoxazole, and azithromycin for *Shigella* and ciprofloxacin and erythromycin for *Campylobacter*. Categorical susceptibilities were recovered from the LIS and/or chart review.

### Demographic, behavioral, and clinical information

Detailed retroactive chart review was performed by two reviewers (T.I.M. and S.B.) to collect demographic information and housing status, symptoms and duration, specialty of evaluating medical provider, results of any stool culture and antibiotic resistance

testing, any documented sexual history, specific sexual behaviors, and history of STIs. Housing status was classified as "housed," "group home," "unstably housed" (no permanent home but living transiently in other people's homes), and "unhoused" (which included shelters and living directly on the streets). Symptoms lasting $\geq 1$ month were classified as chronic (31). Provider specialty was classified as infectious disease/HIV, family medicine, internal medicine, emergency medicine, GI, or other specialty. Whether or not the provider took a sexual history at the presentation of their GI symptoms was recorded as a binary yes/no and was counted as "yes" if any type of sexual history was documented with the visit. Specific sexual behaviors such as penile–oral, penile–anal, and oral–anal were recorded as "yes" if specifically charted the patient engaged in these behaviors. If it was clear from the documentation the patient was not sexually active in the last month, this was documented as "no recent sex"; if it was unclear if the patient was sexually active in the last month, this was documented as "unclear." The number of sexual partners per month was estimated based on the visit note or notes close in date to the patient's BioFire test. Condom use was classified as "yes," "no," "intermittent," or "unclear." Since documentation rarely specified condom usage for a specific sexual behavior, this was not collected. Pre-exposure prophylaxis (PrEP) use was counted as "yes" if the patient had an active prescription and "no" if not. If a patient stated they were not taking prescribed PrEP, this was also counted as "no."

## Statistical methods and data availability

Descriptive and inferential statistics were performed using R Studio Cloud Version 1.4.1717 and GraphPad Prism 9.4.1. Odds ratio (OR), multiple comparison $\chi^2$ test, and Fisher's exact test were all used as statistical tests. The group of patients who were not sexually active in the last month preceding their BioFire test was used as the comparison group for the odds ratio calculations. De-identified data are available upon request.

## RESULTS

### Cohort demographics and pathogens detected

A total of 361 MSM individuals with 436 BioFire tests performed met the inclusion criteria. Of the 361 individuals tested, 238 were persons with HIV (PWH) and 123 were persons without HIV at the time of testing (Table 1). Of the 436 tests, 271 (62%) were positive for at least one pathogen, 165 were negative, and 1 was indeterminate (Fig. 1). Housing status was not significantly associated with a positive or negative result; not included in this analysis due to limited sample size were two individuals who were incarcerated, seven living in their car, and two with an unclear housing status.

In 93.8% of cases, diarrhea was charted as a symptom. Bloody stools (12.6% of cases) and fever (8.4% of cases) had higher, but not statistically significant OR of a positive result compared to diarrhea without these features (Table 1). Most patients presented to the healthcare system within 1 week of symptoms (Fig. 2). Excluding cases of uncertain duration ($N = 17$), 36.8% of patients presented in the first 3 days of symptom onset and 67.9% of patients presented within the first week of symptom onset (Fig. 2).

The six most common pathogens were enteropathogenic *E. coli* (EPEC) (28.7%), enteroinvasive *E. coli* (EIEC) (28.0%), enteroaggregative *E. coli* (EAEC) (22.9%), norovirus (18.9%), *Giardia* (15.5%), and *Campylobacter* (14%). All other pathogens were detected in less than 7% of positive tests (Fig. 3A). Excluding pathogens detected fewer than 10 times, pathogen frequency was significantly different in monomicrobial compared to polymicrobial infections ($\chi^2 = 21.12$, df = 9, $P = 0.0121$). Enterotoxigenic *E. coli* (ETEC) was the pathogen most frequently co-detected with another pathogen (92.8% of ETEC cases), while *Clostridium difficile* was least frequently detected with another pathogen (33% of *C. difficile* cases).

There were 350 acute cases (symptoms for <1 month) and 86 chronic cases. Test positivity in acute cases (67.1%) was significantly higher than for chronic cases (41.9%) (Fisher's exact test, $P < 0.0001$). The most common pathogens in chronic cases were EPEC

**TABLE 1** Demographics and odds ratios of positive results for specific factors[a]

| Demographic | Total | Positive result | Negative result | OR | CI | P value |
|---|---|---|---|---|---|---|
| 361 Unique individuals | | | | | | |
| HIV positive | 238 | | | | | |
| HIV negative | 123 | | | | | |
| Housing status at test | | | | | | |
| Housed | | 213 | 134 | – | – | – |
| Group home | | 15 | 4 | 2.36 | 0.827–6.65 | 0.1491 |
| Unstably housed | | 11 | 10 | 0.692 | 0.299–1.67 | 0.491 |
| Shelter | | 10 | 8 | 0.786 | 0.296–2.00 | 0.6279 |
| Unhoused | | 15 | 5 | 1.89 | 0.660–4.81 | 0.247 |
| HIV status at test | | | | | | |
| Positive | | 194 | 105 | 1.42 | 0.943–2.16 | 0.1098 |
| Negative | | 77 | 59 | – | – | – |
| Charted symptom | | | | | | |
| Diarrhea | | 261 | 148 | – | – | – |
| Blood in stools | | 41 | 14 | 1.66 | 0.892–3.07 | 0.1330 |
| Abdominal pain | | 116 | 62 | 1.06 | 0.738–1.53 | 0.7793 |
| Nausea | | 87 | 47 | 1.05 | 0.703–1.58 | 0.8364 |
| Fever | | 28 | 9 | 1.76 | 0.832–3.88 | 0.2075 |
| Loss of appetite | | 25 | 12 | 1.18 | 0.597–2.36 | 0.7227 |

[a]Unstably housed was defined as not having a permanent home and living periodically at various housed places such as a friend's couch.

($N = 12$), EAEC ($N = 11$), norovirus ($N = 8$), *Shigella*/EIEC ($N = 5$), and *Giardia* spp. ($N = 5$). There were three chronic cases each of *Campylobacter* spp. and *Cryptosporidium* spp., two cases of *C. difficile* and ETEC, and one chronic case of sapovirus and *Yersinia* spp. There was no significant difference in the rate of monomicrobial vs polymicrobial cases in positive acute and chronic cases (Fisher's exact test, $P = 0.4659$).

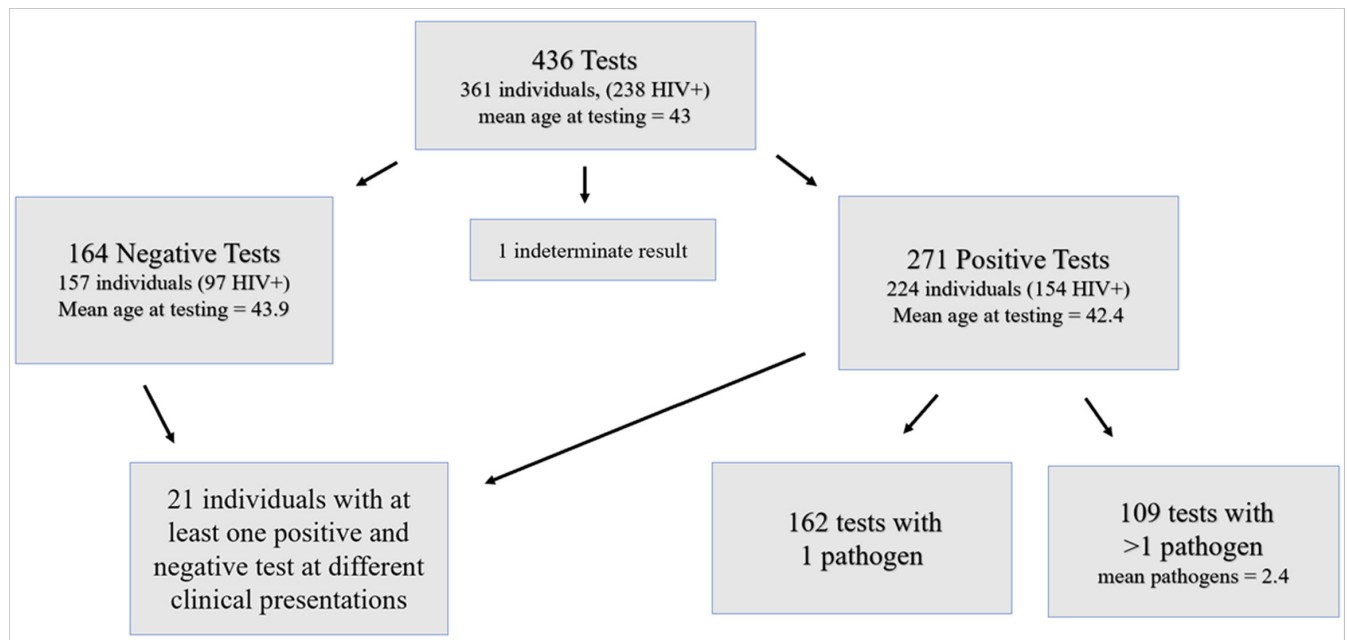

**FIG 1** Diagram of the number of tests and individuals in the analysis. One indeterminate result was excluded from the analysis. A total of 436 tests from 361 individuals were included in the study. For the 57 patients with multiple tests performed, on average they had 2.3 tests, and all multiple tests included were done for different clinical presentations of new symptoms.

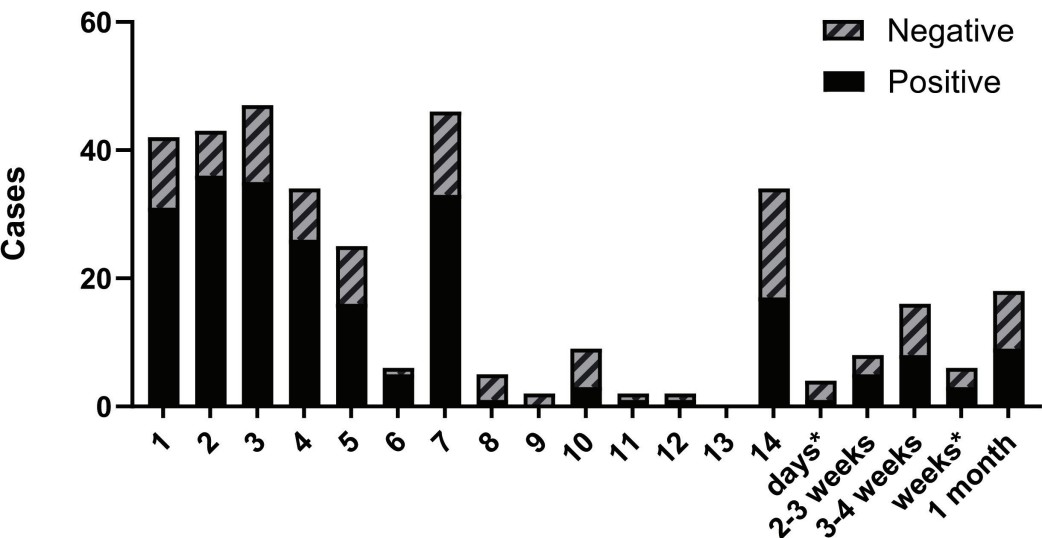

**FIG 2** For cases ≤1 month in duration, the absolute frequency of charted time elapsed from patient's reported symptom onset to healthcare presentation. The duration that was >14 days but less than ≤1 month was combined into week or month categories. *In some instances, the visit notes stated the duration of symptoms as "days" or "weeks" rather than a more specific time frame.

BioFire cannot discriminate between *Shigella* and EIEC; however, reflexed culture was performed on all cases (*N* = 76) and was positive for *Shigella flexneri* in 47 cases, *S. sonnei* in 6 cases, and *Shigella* not otherwise specified (NOS) in 3 cases (Table 2). In 20 *Shigella/*EIEC-positive cases, neither organism grew in culture; EIEC did not grow in any reflex cultures. There was nearly 100% resistance of *Shigella* spp. isolates to ampicillin except for one case of susceptible *S. sonnei*. Most isolates were also resistant to trimethoprim/sulfamethoxazole (86%) and azithromycin (82%), but sensitive to ciprofloxacin (91%; Table 2). Three cases of *S. flexneri* and one case of *Shigella sonnei* were resistant to ciprofloxacin, trimethoprim/sulfamethoxazole, azithromycin, and ampicillin. However, all four of these cases were sensitive to ceftriaxone.

Reflex culture in *Campylobacter*-positive cases grew *C. jejuni* in 13 cases, *C. coli* in one case, and *Campylobacter* NOS in one case; in 23 *Campylobacter*-positive cases, it did not grow in reflex culture. All 13 cases of *C. jejuni* were resistant to ciprofloxacin, and one case was also resistant to erythromycin. One case of *Campylobacter* NOS was resistant to ciprofloxacin and also resistant to ertapenem (erythromycin was not tested in this case). One case of *C. coli* was sensitive to both ciprofloxacin and erythromycin (Table 2).

## HIV-specific findings

The positivity rate for tests from PWH at the time of testing was 194/300 (65%) compared to 77/136 (57%) for tests from persons without HIV. Living with HIV was not statistically associated with a positive result (OR = 1.40, 95% CI = 0.935–2.14, *P* = 0.1114). For persons without HIV, PrEP use was not statistically significant (Table 3). *Shigella* was significantly more likely for PWH (*P* = 0.0247), while *Giardia* was significantly more likely in persons without HIV (*P* = 0.0047) (Fig. 3B). Among pathogens detected two times or more, rotavirus and astrovirus were only identified in PWH. Viral load was undetectable (<40 copies/mL) or suppressed (≤200 copies/mL) in 86% of PWH at the time of viral load testing closest to enteric pathogen testing. Unsuppressed viral load (>200 copies/mL) had no impact on overall positivity (OR = 1.05, 95% CI 0.51–2.11, *P* > 0.999). Among pathogens with eight cases or more, *Cryptosporidium* was significantly more common in individuals with an unsuppressed viral load (14.6% vs 3.1% of suppressed, *P* = 0.0059).

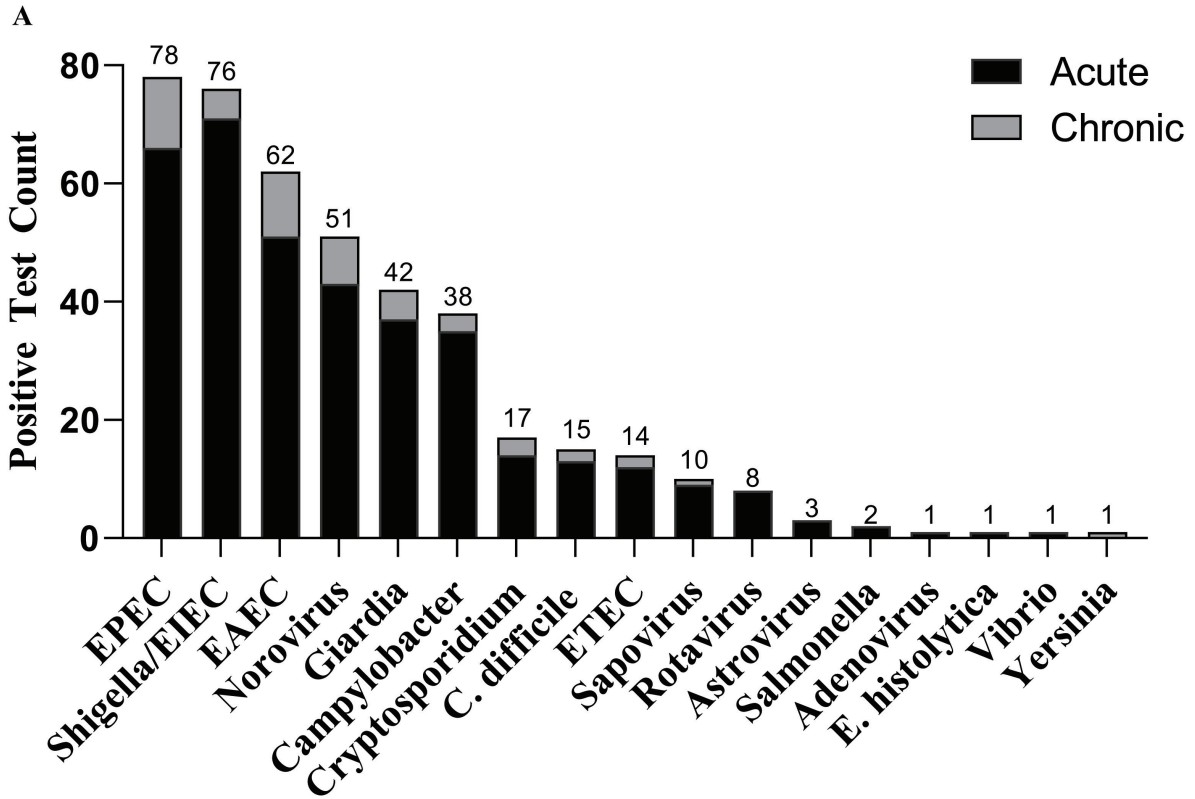

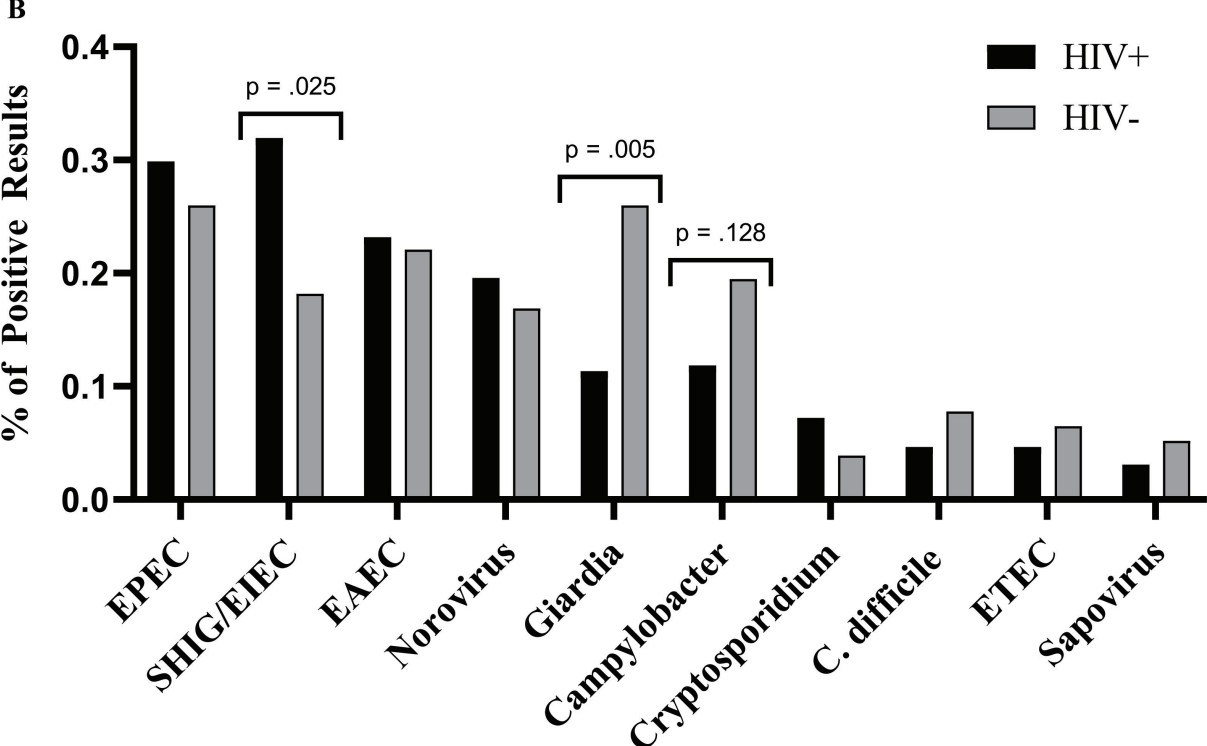

**FIG 3** (A) Absolute frequency of each pathogen identified and comparison of pathogen absolute frequency between acute cases (<1 month of symptoms) vs chronic cases (≥1 month of symptoms). Bar labels indicate total tests positive for the organism indicated. (B) Comparison of pathogen frequency between individuals living with and without HIV.

**TABLE 2** Rates of antibiotic resistance to tested antibiotics in isolates from reflexive cultures

| Shigella | Cases | Ampicillin | Trimethoprim/sulfamethoxazole | Azithromycin (E-test >2) | Ciprofloxacin |
|---|---|---|---|---|---|
| *Shigella flexneri* | 47 | 47 (100%) | 39 (83%) | 37 (79%) | 4[a] (9%) |
| *S. sonnei* | 6 | 5 | 6 | 6 | 1 |
| *Shigella* NOS | 3 | 3 | 3 | 3 | 0 |
| **Campylobacter** | **Cases** | **Ciprofloxacin** | **Erythromycin** | | |
| *Campylobacter jejuni* | 13 | 13 (100%) | 1 (8%) | | |
| *C. coli* | 1 | 0 | 0 | | |
| *Campylobacter* NOS | 1 | 1 | (Not done) | | |

[a]One case of *S. sonnei* and three of the four cases of *S. flexneri* with ciprofloxacin resistance were also resistant to ampicillin, trimethoprim/sulfamethoxazole, and azithromycin, but all of these cases were sensitive to ceftriaxone. One case of *S. flexneri* was resistant to ampicillin, azithromycin, and ciprofloxacin but was sensitive to both trimethoprim/sulfamethoxazole and ceftriaxone.

## Sexual history

Providers documented a sexual history concurrently in the encounter associated with testing in 177 (40.6%) of the 436 tests performed. Of the 271 positive tests, providers charted a sexual history in 104 (38.3%) of these instances. Provider type significantly differed in how often sexual history was charted ($x^2$ = 33.74, df = 5, $P \leq$ 0.0001). Specifically, HIV/infectious disease providers charted sexual behavior with the visit 53.8% of the time, internal medicine 41.2%, family medicine 33%, emergency medicine 22.1%, and other specialty providers 22.2%.

Patient sexual activity near the time of testing was documented in 331 tests (201 positive cases and 130 negative cases). Any reported sexual activity was significantly more likely to have a positive result compared to those not currently sexually active. Anilingus (insertive/receptive not specified) had the highest OR of a positive result (OR = 43.2, $P \leq$ 0.0001) followed by penile–anal (insertive/receptive not specified) sex (OR = 5.63, $P \leq$ 0.0001) (Table 3).

Patients reporting more sexual partners per month were also significantly more likely to have a positive result compared to those not currently sexually active. The highest ORs were observed for patients reporting two to five partners per month (OR = 11.4, $P \leq$ 0.0001) and multiple partners per month (OR = 11.1, $P \leq$ 0.0001). Patients who reported only one sexual partner had the lowest risk (OR = 3.26, $P$ = 0.0080) (Table 3). The number of sexual partners was unclear in 81 positive cases and 48 negative cases. Among 221 cases where condom use could be ascertained, no condom use and intermittent condom use were significantly more likely to have a positive result compared to those who reported consistent condom usage (Table 3).

In 61% ($N$ = 266) of tests, the individual had a history of gonorrhea or chlamydia in the last 3 years or any prior history of syphilis. Among the 271 positive enteric pathogen tests, 66% had a prior STI vs 54% of the negative enteric pathogen results. Additionally, in 20 instances of a positive enteric pathogen test (7.3% of positive tests), the patient was concurrently diagnosed with either rectal gonorrhea ($N$ = 10) or rectal chlamydia ($N$ = 10). Six individuals with a negative enteric pathogen test (3.7% of negative test results) were diagnosed with rectal gonorrhea ($N$ = 3) or rectal chlamydia ($N$ = 3). Overall, concurrent rectal chlamydia or gonorrhea had an elevated but not significant risk for a positive enteric pathogen result (OR = 2.19, $P$ = 0.1016).

## DISCUSSION

Our study supports the observation that enteric pathogens are likely important sexually transmitted infections in the MSM population. The BioFire positivity rate in this MSM cohort (62%) was nearly double the previously reported overall positivity rate from our institution of 33.5%, but similar to the positivity rate of MSM cohorts in other studies (52.5%–56.3%) (28, 29, 32). Sexual activity had a higher OR of a positive result, agreeing with other recent studies (25, 26, 33, 34). Specifically, concordant with another study,

**TABLE 3** Sexual factors with odds ratios

| Risk factor | Positive | Negative | OR | 95% OR CI | P value |
|---|---|---|---|---|---|
| Sexual partners | | | | | |
| 0 | 22 | 50 | – | – | – |
| 1 | 43 | 30 | 3.26 | 1.61–6.60 | 0.0080 |
| ≥1[a] | 18 | 12 | 3.41 | 1.40–8.12 | 0.0076 |
| 2–5 | 25 | 5 | 11.4 | 3.69–29.4 | <0.0001 |
| 6+ | 9 | 4 | 5.11 | 1.49–16.0 | 0.0118 |
| Multiple[b] | 73 | 15 | 11.1 | 5.28–22.5 | <0.0001 |
| Sexual behavior | | | | | |
| No recent sex | 22 | 50 | – | – | – |
| Oral–penile | 103 | 59 | 3.97 | 2.15–7.17 | <0.0001 |
| Penile–anal sex | | | | | |
| Receptive (strictly) | 34 | 19 | 4.07 | 1.88–8.60 | 0.0003 |
| Insertive (strictly) | 20 | 11 | 4.13 | 1.71–10.2 | 0.0020 |
| Versatile | 105 | 33 | 7.09 | 3.71–13.1 | <0.0001 |
| Overall[c] | 161 | 65 | 5.63 | 3.16–9.86 | <0.0001 |
| Anilingus (rimming) | | | | | |
| Insertive | 18 | 1 | 40.9 | 5.92–435.1 | <0.0001 |
| Receptive | 7 | 0 | [d] | [d] | [d] |
| Unspecified | 16 | 1 | 36.4 | 6.03–389 | <0.0001 |
| Overall | 38 | 2 | 43.2 | 10.0–188 | <0.0001 |
| Any reported sex | 170 | 74 | 5.22 | 2.98–9.02 | <0.0001 |
| Reported monogamy | 13 | 10 | 2.96 | 1.13–7.2 | 0.0453 |
| Condom use | | | | | |
| Yes | 7 | 10 | – | – | – |
| No | 76 | 25 | 4.02 | 1.34–11.0 | 0.0151 |
| Intermittent | 73 | 30 | 3.03 | 1.04–8.04 | 0.0477 |
| PrEP use | | | | | |
| No | 36 | 33 | – | – | – |
| Yes | 41 | 26 | 1.45 | 0.748–2.85 | 0.305 |
| Prior STI[e] | | | | | |
| No | 92 | 78 | - | - | - |
| Syphilis | 129 | 55 | 1.99 | 1.28–3.06 | 0.0021 |
| *Neisseria gonorrhea* | | | | | |
| Rectal | 62 | 24 | 2.19 | 1.28–3.87 | 0.0068 |
| Pharyngeal | 65 | 26 | 2.12 | 1.22–3.62 | 0.0079 |
| Urethral | 29 | 8 | 3.07 | 1.39–6.67 | 0.0093 |
| Overall[f] | 108 | 49 | 1.87 | 1.20–2.94 | 0.0089 |
| *Chlamydia trachomatis* | | | | | |
| Rectal | 64 | 33 | 1.65 | 0.999–2.73 | 0.0707 |
| Pharyngeal | 27 | 10 | 2.25 | 1.03–4.83 | 0.0434 |
| Urethral | 19 | 12 | 1.34 | 0.620–2.98 | 0.5570 |
| Overall[f] | 88 | 43 | 1.74 | 1.07–2.80 | 0.0245 |

[a]"≥1" means it cannot be determined from the chart if the patient either had only one sex partner or multiple sex partners as opposed to "1" where the patient had only one sex partner.
[b]Multiple partner category means the patient reported more than one sex partner but a more specific number was not charted.
[c]Overall penile–anal sex category also includes two cases where the type of penile–anal sex was not specified.
[d]An OR cannot be calculated because there were zero negative cases for receptive anilingus.
[e]*N. gonorrhea* and *C. trachomatis* cases were included in this table if the test date was within the last three prior years before the date of the BioFire test and not performed concurrently with BioFire testing. Any prior history of syphilis infection was included.
[f]Overall cases represent each case that had at least one site (rectal, pharyngeal, or urethral) positive for the organism (many cases had a history of more than one positive site) and also includes prior history that did not specify the site of testing.

anilingus in our study was by far the most significant risk factor, which is not surprising given that enteric pathogens follow fecal–oral transmission (26).

The most commonly identified pathogens in this study were EAEC, EPEC, *Shigella*/EIEC, *Giardia*, *Campylobacter*, and norovirus, which have also been identified as frequent pathogens in other recent MSM enteric pathogen studies utilizing multiplex PCR (25, 29, 32). EPEC and EAEC were monopathogens in 32.1% of cases positive for EPEC and 30.6% of cases positive for EAEC, and specifically, the chronic cases largely consisted of EPEC and EAEC over other pathogens. This adds to the literature that EPEC and EAEC are pathogenic (35–40).

Concordant with other recent studies, we identified several MDR *Shigella* spp. isolates (10–12, 15, 41). All *Campylobacter* spp. isolates were resistant to ciprofloxacin except for one case of *C. coli*, consistent with predicted resistance. Every *Campylobacter* isolate was sensitive to erythromycin except for one *C. jejuni* (42).

Living with or without HIV had no significant difference in overall positivity, concordant with two other recent studies (29, 33). *Shigella* spp. and *Giardia* detection rates did significantly differ; *Shigella* was found more frequently in PWH, while *Giardia* was found more frequently in persons without HIV. The reason for higher *Giardia* rates specifically in PWH in this study is not clear, although one possible factor may be serosorting—the behavior where PWH only have sex with other PWH and those without HIV only have sex with others who do not have HIV (23, 43, 44).

Some studies have demonstrated *Shigella* to be more prominent in PWH (17, 45). This may be due to HIV-related increased susceptibility to *Shigella* and/or serosorting (46). In our study, immunodeficiency is unlikely to account for the increased risk of shigellosis since an undetectable viral load was not protective compared to PWH who had a quantifiable viral load, in agreement with a 2018 study (45). Serosorting or some other unknown cause may be a better explanation for the difference.

The use of PrEP was investigated as a risk factor because PrEP has been associated with increased condomless anal intercourse, and this may increase the risk of fecal–oral transmission (47). In this study, PrEP was not significantly associated with an increased positive test. The literature on this has been conflicting, and sample size and the presence of symptoms in the cohort may be confounding factors (26, 33).

This study only captured patients who were identified in the chart as MSM and also missed patients for whom testing was never ordered and/or who never presented for medical care. Thus, the true rate of enteric pathogens within MSM may be overall different. These factors were also likely why there were more PWH in this study compared to the overall rate of HIV infection of ~17% within the MSM population (48). In addition, HIV providers may have been more likely to test for infectious causes of diarrhea in their MSM patients.

Some providers may have asked specifically about anilingus when they had a higher suspicion of an infectious cause of the GI illness, which would artificially inflate the OR of anilingus for a positive result. It is also possible that the increased OR of other sexual behaviors was confounded by anilingus that occurred concurrently but was not reported/documented.

Access to a hygienic environment was not an important factor for transmission in this MSM cohort as housing status did not correlate with the risk of a positive result. This finding is concordant with Tansarli et al. (49), who using whole-genome sequencing, identified discrete clonal lineages within an MSM population and a population of unhoused persons, indicating that transmission was largely separate between the two populations (49). Providers need to be aware that MSM are at a higher risk for enteric pathogens regardless of their housing status.

Alarmingly, this study suggests that provider vigilance of enteric STI risk in the MSM population is lacking. Overall providers did not chart a sexual history in the majority of visits, and while there may be other reasons for this besides lack of awareness, it raises the question if it is predominantly because of a lack of education. Enteric STI risk within the MSM population has been known since the 1970s, yet providers may not be aware (6). Our study also demonstrates that provider recognition of enteric pathogens

as STIs could help with case detection for traditionally recognized STIs—like syphilis and gonorrhea—at the time of enteric pathogen testing.

MSM community knowledge of enteric STI risk is low. In the United Kingdom, a large 2017 survey of MSM found that only 26.6% had heard of *Shigella* and only 16.5% knew it was an STI (50). Many patients in our study waited a week or more to present for clinical care, and while there are many possible reasons for this such as challenging access to care, this could partially be from a lack of awareness of STI risk. Fostering provider awareness should help facilitate patient awareness.

Community interventions to increase MSM awareness of enteric STIs are needed. One MSM focus group relayed they would prefer these interventions to demonstrate people of various orientations to reduce stigma (51). Ultimately, enteric STIs are not limited to the MSM population, and any individual who engages in sexual activity that involves fecal–oral transmission may be at an increased risk regardless of sexual orientation or their partner's genitalia. Anal intercourse is not an uncommon behavior within the heterosexual community, which could result in fecal–oral contamination depending on the other behaviors associated with it (52). It may be that enteric STI transmission is being largely underrecognized in other cohorts besides MSM.

## ACKNOWLEDGMENTS

We thank Drs. Ferric Fang and Gianna Tansarli for helpful discussions of the study and its findings, as well as Dr. Patrick Mathias for help constructing the natural language search of the EMR for inclusion criteria.

## AUTHOR AFFILIATIONS

[1]Department of Laboratory Medicine and Pathology, University of Washington, Seattle, Washington, USA
[2]Department of Internal Medicine, Brigham and Women's Hospital, Boston, Massachusetts, USA

## AUTHOR ORCIDs

Timothy Isaac Miller http://orcid.org/0009-0007-4733-4728
Joshua A. Lieberman http://orcid.org/0000-0002-0376-1337

## AUTHOR CONTRIBUTIONS

Timothy Isaac Miller, Conceptualization, Data curation, Formal analysis, Investigation, Methodology, Project administration, Resources, Software, Supervision, Writing – original draft, Writing – review and editing | Stephanie Banning, Data curation, Investigation, Writing – review and editing | Joshua A. Lieberman, Investigation, Methodology, Resources, Supervision, Writing – original draft, Writing – review and editing

## ETHICS APPROVAL

The study was approved by the Institutional Review Board at the University of Washington (STUDY00014288).

## ADDITIONAL FILES

The following material is available online.

### Open Peer Review

**PEER REVIEW HISTORY (review-history.pdf).** An accounting of the reviewer comments and feedback.

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
