## [Reviewer comments · Microbiology Spectrum]

Microbiology Spectrum

Risk Factors and Provider Awareness of Sexually Transmitted Enteric Pathogens among Men Who Have Sex with Men

Timothy Miller, Stephanie Banning, and Joshua Lieberman

Corresponding Author(s): Timothy Miller, University of Washington

Review Timeline:

Submission Date:	October 17, 2023
Editorial Decision:	November 16, 2023
Revision Received:	January 8, 2024
Accepted:	January 30, 2024

Editor: Vera Tesic

Reviewer(s): Disclosure of reviewer identity is with reference to reviewer comments included in decision letter(s). The following individuals involved in review of your submission have agreed to reveal their identity: Aniruddha Hazra (Reviewer #1); Kanan T Desai (Reviewer #2)

Transaction Report:

DOI: <https://doi.org/10.1128/spectrum.03577-23>

Re: Spectrum03577-23 (Risk Factors and Provider Awareness of Sexually Transmitted Enteric Pathogens among Men Who Have Sex with Men)

Dear Dr. Timothy Isaac Miller:

Thank you for the privilege of reviewing your work. Below you will find my comments, instructions from the Spectrum editorial office, and the reviewer comments.

Revision Guidelines

Sincerely,
Vera Tesic
Editor
Microbiology Spectrum

Reviewer #1 (Comments for the Author):

This is a very interesting study looking at factors associated with GI BioFire positivity among MSM at a large academic hospital.

General Comment

Please revise manuscript to incorporate person first language throughout. We should no longer identify patients or participants

as HIV-positive or HIV-infected. They are people with HIV (PWH) or people living with HIV (PLWH). Similarly, would change HIV-negative to people without HIV.

Methods

How did you define PrEP use? Do you mean active PrEP prescription at time of testing on chart review? These are two different things

Please include how sexual behaviors were collected on chart abstraction. For example, for oral-anal contact, if nothing was documented, was the patient considered to have no oral-anal contact? How did statistical analysis account for this?

Results

To follow results better, consider documenting % and absolute numbers

Line 173 - HIV specific findings, 194 positive out of 300 PWH tested; however earlier it states that there were 238 PWH in the cohort. Similar discrepancy for people without HIV, please clarify

Line 174-177 - Would just say that the small absolute increases in both groups were not statistically significant.

Line 177-179 - Remove this sentence and discuss statistically significant organisms (Shigella and Giardia)

Line 184 - Why is >40 c/mL used to define unsuppressed VL? This number is not clinically meaningful; would consider >200 c/mL

Line 209 - What is the clinical significance of 3 years for an STI to be consider "recent"? Most guidelines and data, specifically in MSM, define recent as 12 months

Discussion

Line 218 - While I agree with this statement, I am not sure your data fully supports this. You present retrospective data of MSM with positive enteric pathogen screening. Sexual history data is somewhat incomplete or remote, of what is documented there is a correlation of higher sexual activity and oral-anal contact with positivity of tests.

Line 242 - Serosorting practices have changed dramatically over the past 10 years with expansion of PrEP and U=U messaging, not sure this argument holds water currently

Reviewer #2 (Comments for the Author):

The discussion is very long and can be truncated.

In the methods section, details on how samples for the Biofire test are collected can be added.

It will also be important to add some background information on Biofire, e.g., what is the test? What is sensitivity specificity, etc?

We thank the reviewers for their time and attention. Below, please find our point-by-point responses and descriptions of our modifications to the manuscript.

Reviewer 1 (Comments for the author):

General Comment

Please revise manuscript to incorporate person first language throughout. We should no longer identify patients or participants as HIV-positive or HIV-infected. They are people with HIV (PWH) or people living with HIV (PLWH). Similarly, would change HIV-negative to people without HIV.

Author response: Thank you for educating on this. This language was changed throughout the paper.

Methods

How did you define PrEP use? Do you mean active PrEP prescription at time of testing on chart review? These are two different things.

Author response: We have added text (lines: 131-133, revised manuscript) clarifying that PrEP use was defined as “yes” if the patient had an active prescription. Patients were categorized as “no PrEP” if the patient either did not have a prescription or if, per review of clinical notes, they reported not taking their prescription.

Please include how sexual behaviors were collected on chart abstraction. For example, for oral-anal contact, if nothing was documented, was the patient considered to have no oral-anal contact?

Author response: We have added clarifying text (lines 124-128, revised manuscript). Patients were documented as “yes” for a specific sexual behavior if and only if it was charted that they engaged in this sexual behavior. If it could be determined the patient had not been sexually active in the last month, this was documented as “no recent sex.”

How did statistical analysis account for this?

Author response: Statistical analysis used odds ratio of a positive result versus negative results, comparing specific sexual behaviors to the group with no recent sex in the last month (lines 140-142, revised manuscript).

Results

To follow results better, consider documenting % and absolute numbers

Line 173 - HIV specific findings, 194 positive out of 300 PWH tested; however earlier it states that there were 238 PWH in the cohort. Similar discrepancy for people without HIV, please clarify

Author response: This is because there are 238 people living with HIV but 300 tests from people with HIV. Some patients had multiple tests, more likely in PWH. We have updated language to better clarify this (lines 192-193, revised manuscript).

Line 174-177 - Would just say that the small absolute increases in both groups were not statistically significant.

Author response: We have made the recommended change.

Line 177-179 - Remove this sentence and discuss statistically significant organisms (Shigella and Giardia).

Author response: We have made the recommended change.

Line 184 - Why is >40 c/mL used to define unsuppressed VL? This number is not clinically meaningful; would consider >200 c/mL

Author response: We originally used the reportable range of the clinical assay used in our hospital system to define unsuppressed viral load. We have recalculated using the recommended threshold of >200 c/mL. Updated results appear in lines 198-203 (revised manuscript).

Line 209 - What is the clinical significance of 3 years for an STI to be consider "recent"? Most guidelines and data, specifically in MSM, define recent as 12 months

Author response: We selected three years was used partially because our study overlapped with COVID, during which STI diagnoses fell likely reflecting under-reporting and under-diagnosis, and were thus different than pre-pandemic rates (<https://www.cdc.gov/std/statistics/2021/impact.htm>). We therefore elected to capture prior history of STIs and expanded the window to 3 years.

Discussion

Line 218 - While I agree with this statement, I am not sure your data fully supports this. You present retrospective data of MSM with positive enteric pathogen screening. Sexual history data is somewhat incomplete or remote, of what is documented there is a correlation of higher sexual activity and oral-anal contact with positivity of tests.

Author response: We have revised the language accordingly to be less conclusive. We now state: “Our study supports the observation that enteric pathogens are likely important sexually transmitted infection in the MSM population.” (Lines 234-235, updated manuscript).

Line 242 - Serosorting practices have changed dramatically over the past 10 years with expansion of PrEP and U=U messaging, not sure this argument holds water currently.

Author response: We have tempered the language to indicate serosorting may be a *possible / contributory* factor rather than sole explanation. We have also added a recent citation that shows serosorting is still occurring.

Wang L et al. Population-Level Sexual Mixing According to HIV Status and Preexposure Prophylaxis Use Among Men Who Have Sex With Men in Montreal, Canada: Implications for HIV Prevention. *Am J Epidemiol.* 2020 Jan 31;189(1):44-54. doi: 10.1093/aje/kwz231. PMID: 31612213.

Reviewer #2 (Comments for the Author):

The discussion is very long and can be truncated.

Author responses: We have cut out text where possible to streamline the discussion.

In the methods section, details on how samples for the Biofire test are collected can be added. It will also be important to add some background information on Biofire, e.g., what is the test? What is sensitivity specificity, etc?

Author response: We have added this information in methods, lines 93-95 (revised manuscript). We state “Biofire is a rapid multiplex GI PCR test performed on a stool sample in Cary Blair medium transported to the laboratory within 72 hours. The assays tests for twenty-two different pathogens with a sensitivity of 94.5-100% and a specificity of 97.1-100% depending on the target.”

Re: Spectrum03577-23R1 (Risk Factors and Provider Awareness of Sexually Transmitted Enteric Pathogens among Men Who Have Sex with Men)

Dear Dr. Timothy Isaac Miller:

Your manuscript has been accepted, and I am forwarding it to the ASM production staff for publication. Your paper will first be checked to make sure all elements meet the technical requirements. ASM staff will contact you if anything needs to be revised before copyediting and production can begin. Otherwise, you will be notified when your proofs are ready to be viewed.

Sincerely,
Vera Tesic
Editor
Microbiology Spectrum

Reviewer #1 (Comments for the Author):

Thank for addressing all comments

Reviewer #2 (Comments for the Author):

Overall manuscript is improved from last version. It could still be made more comprehensive and avoiding repetition.